# Large optical nonlinearity of ITO nanorods for sub-picosecond all-optical modulation of the full-visible spectrum

Peijun Guo[1], Richard D. Schaller[2,3], Leonidas E. Ocola[2], Benjamin T. Diroll[2], John B. Ketterson[4] & Robert P.H. Chang[1]

Nonlinear optical responses of materials play a vital role for the development of active nanophotonic and plasmonic devices. Optical nonlinearity induced by intense optical excitation of mobile electrons in metallic nanostructures can provide large-amplitude, dynamic tuning of their electromagnetic response, which is potentially useful for all-optical processing of information and dynamic beam control. Here we report on the sub-picosecond optical nonlinearity of indium tin oxide nanorod arrays (ITO-NRAs) following intraband, on-plasmon-resonance optical pumping, which enables modulation of the full-visible spectrum with large absolute change of transmission, favourable spectral tunability and beam-steering capability. Furthermore, we observe a transient response in the microsecond regime associated with slow lattice cooling, which arises from the large aspect-ratio and low thermal conductivity of ITO-NRAs. Our results demonstrate that all-optical control of light can be achieved by using heavily doped wide-bandgap semiconductors in their transparent regime with speed faster than that of noble metals.

[1] Department of Materials Science and Engineering, Northwestern University, 2220 Campus Drive, Evanston, Illinois 60208, USA. [2] Center for Nanoscale Materials, Argonne National Laboratory, 9700 South Cass Avenue, Lemont, Illinois 60439, USA. [3] Department of Chemistry, Northwestern University, 2145 Sheridan Road, Evanston, Illinois 60208, USA. [4] Department of Physics and Astronomy, Northwestern University, 2145 Sheridan Road, Evanston, Illinois 60208, USA. Correspondence and requests for materials should be addressed to R.D.S. (email: schaller@anl.gov) or to R.P.H.C. (email: r-chang@northwestern.edu).

When exposed to an ultrafast laser pulse, the distribution of conduction band electrons in metals can be modified to cause changes in the real and imaginary parts of the permittivity that are manifested optically[1]. In particular, nonlinear plasmonic response arising from the coupling of intense electromagnetic fields to the conduction band electrons in metallic nanostructures can provide all-optical modulation with sizable on-off ratios and high speeds, which may provide a viable way to integrate optical signal switching and processing elements into optical networks[2–5]. Noble metallic nanostructures possess large negative permittivity in the visible and near-infrared range, and can therefore concentrate optical fields into subwavelength dimensions[6] with enhanced nonlinear plasmonic response[7–11]. However, the high electron concentration in noble metals limits the extent to which the electron distribution can be modified and with it the achievable permittivity modulation. In addition, the strong interband transitions in the visible range (such as those from the d-band to the Fermi-surface in gold at an energy of ~2.4 eV) give rise to a large dispersion of the permittivity modulation versus wavelength, which furthermore can overlap with their plasmonic resonances, thereby complicating the design of nonlinear optical devices. Graphene as an emerging active plasmonic material has attracted significant attention as it offers ultrafast response in the sub-picosecond regime[12], and can be further interfaced with other two-dimensional atomic crystals for dispersion engineering[13]; however, the operation wavelength of graphene based plasmonics is primarily limited to the mid-infrared range. The utilization of dielectrics and metals that respond to light in drastically different ways has brought novel schemes for manipulating visible light at subwavelength scales[14–18]. Transparent conducting oxides (TCOs), in particular, are metallic in the near-infrared to mid-infrared[19–26] but become dielectric in the visible, which raises the intriguing question of whether novel nanophotonic functionalities can be achieved if both the metallic and dielectric properties of TCO materials are simultaneously exploited.

Here we demonstrate large optical nonlinearity of indium tin oxide nanorod arrays (ITO-NRAs) in the dielectric range from 360 to 710 nm (denoted as the visible range) when pumped at the localized surface plasmon resonance (LSPR) in the near-infrared. A number of transmission minima in the visible range, arising from collective light diffraction by the periodic dielectric nanorod array, gives rise to a pump-induced modulation of transmission with absolute amplitude up to ±20%. These transmission minima also act as sensitive 'probes' for the quantification of permittivity change and thereby the optical nonlinearity of ITO. We show that a positive change of the real part of the permittivity is achieved throughout the visible range, which is attributed to a modification of the interband transitions in ITO. Moreover, the large scattering cross-section of the dielectric ITO-NRAs (as opposed to the large absorption cross-section of noble metallic nanostructures) allows for a dynamic redistribution of light intensities among different diffraction orders, and the spectral response of the ITO-NRAs can be tuned by simply adjusting the incidence angle or tailoring the length of the nanorods. In the temporal domain, we found both a sub-picosecond response stemming from the electron-phonon coupling and a microsecond response arising from the lattice cooling in ITO.

## Results

### Static spectral features of the ITO-NRA.
Figure 1a shows a scanning electron microscopy image of the highly uniform ITO-NRA achieved by the vapour-liquid-solid growth process (see Methods). Static transmission spectra of the array in the visible and near-infrared range, both measured at normal incidence, are shown in Fig. 1b,c, respectively. The transmission suppression centred at 1,500 nm, shown in Fig. 1c is due to strong light absorption of the ITO-NRA at its transverse LSPR with electrons oscillating along the short axis[20]. The visible spectrum shown in Fig. 1b, exhibits five pronounced transmission minima centred at 589, 486, 434, 396 and 369 nm (denoted as $\lambda_1$, $\lambda_2$, $\lambda_3$, $\lambda_4$ and $\lambda_5$), respectively. The transmission minima in the visible regime are not due to resonant absorption[27] but are simply standing wave resonances based on the $HE_{11}$ waveguide mode. As illustrated by the near-field plots shown in Fig. 1d obtained from optical simulations, at each wavelength of transmission minimum the waves reach an out-of-phase condition at the interface of the nanorod and substrate (which is at the bottom boundary of the nanorod)[28]. The out-of-phase condition is mathematically represented as a phase difference equal to $(2m-1)\pi$, with $m=1$, 2, 3, 4 and 5 corresponding to the transmission minimum at $\lambda_1$, $\lambda_2$, $\lambda_3$, $\lambda_4$ and $\lambda_5$, respectively. By treating each nanorod as a dielectric waveguide supporting the fundamental nanorod $HE_{11}$ mode (see Supplementary Fig. 1 and Supplementary Note 1), the spectral locations of the transmission minima can be related to the effective mode index (denoted as $n_{eff}$) of the waveguide by the equation $2\pi(h \cdot n_{eff}/\lambda_m - h \cdot 1/\lambda_m) = (2m-1)\pi$, or equivalently, $\lambda_m = 2h(n_{eff}-1)/(2m-1)$, here denoted as equation (1), where $\lambda_m$ is the wavelength of a transmission minimum and $h$ is the height of the nanorod. Fig. 1e depicts the wavelength dependence of $n_{eff}$ predicted by the waveguide simulations, which matches well with the $n_{eff}$ calculated from equation (1) using the experimentally observed wavelengths of the transmission minima. The permittivity of ITO used in both the optical and waveguide simulations in the visible range is based on the Drude–Lorentz model with parameters obtained from matching the experimentally measured transmission spectrum by the simulated analogue (see Supplementary Fig. 2 to Fig. 3, and Supplementary Note 2).

While the near-infrared LSPR is a localized phenomenon, the transmission minima in the visible range are due to coherent light diffraction by the ITO-NRA and therefore is attributed to an array effect. Effectively, the ITO-NRA acts as a two-dimensional diffraction grating that supports not only the forward propagating (0, 0) order, but also the (1, 0) and (1, 1) grating orders propagating in oblique directions (details in Supplementary Note 1 and Supplementary Fig. 1). The dielectric nature of ITO in the visible range dictates that intensities of the (1, 0) and (1, 1) grating orders should be complementary to that of the (0, 0) order. This is supported by the transmission and reflection intensities of the ITO-NRA measured using an integrating sphere (spectra shown in Supplementary Fig. 4 and Supplementary Note 3), and is further confirmed by the transmission spectra of the higher diffraction orders plotted in Fig. 1f obtained from optical simulations; the transmission maxima of the (1, 0) and (1, 1) orders are found to spectrally match the transmission minima of the (0, 0) order.

### Transient absorption experiments on the ITO-NRA.
The transient spectral response of the ITO-NRA was investigated by pump-probe transient absorption (TA) experiments. To fully characterize the dynamics, we performed both nanosecond TA experiments (denoted as short-delay-TA experiments) and microseconds TA experiments (denoted as long-delay-TA experiments). In both experiments the centre wavelength of the pump was tuned to the LSPR wavelength of 1,500 nm, which permits large on-resonance absorption in the metallic region of the ITO-NRA. Pumping the sample at 800 nm (off-resonance) was found to give significantly weaker response in comparison to

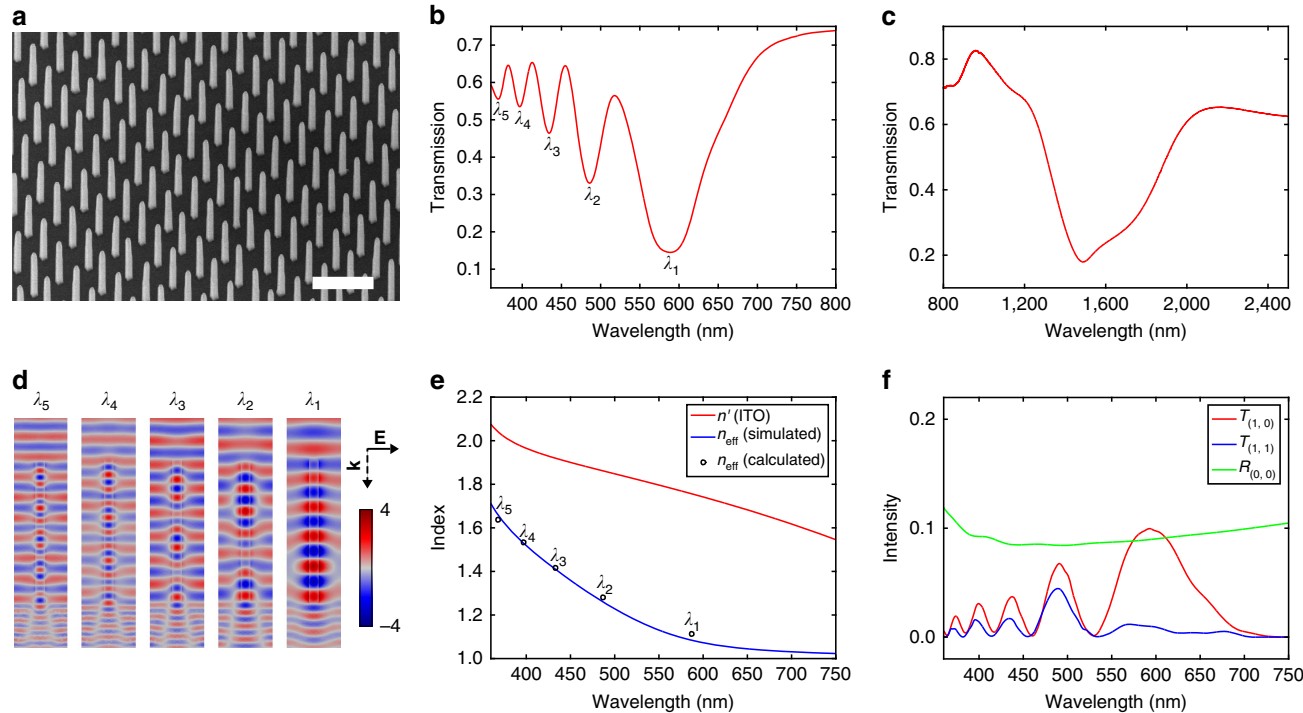

**Figure 1 | Experimental and simulated spectral features of the ITO-NRA in the static case.** (**a**) 30° tilted Scanning Electron Microscopy (SEM) image of the ITO-NRA (1 μm pitch size, 2.6 μm height, 180 nm side length). Scale bar, 2 μm. (**b**) Experimental static transmission spectrum in the visible range (referenced to air). $\lambda_1$ to $\lambda_5$ indicate the wavelengths of the transmission minima. (**c**) Experimental static transmission spectrum in the near-infrared range (referenced to air). (**d**) Near-field plots for the five transmission minima marked in **b**. Electric field intensities are plotted for the plane cutting through the centre of the nanorod, and are normalized by the electric field intensity of the incident wave. (**e**) Refractive index of ITO, $n'$, and the simulated effective mode index of the nanorod waveguide, $n_{eff}$. Circles are the mode index calculated by equation (1) based on experimentally observed transmission minima wavelengths. (**f**) Simulated transmission spectra for one of the four equivalent (1, 0) and (1, 1) grating orders, shown as $T_{(1, 0)}$ and $T_{(1, 1)}$, and the zero degree (0, 0) reflection (shown as $R_{(0, 0)}$).

the on-resonance pumping (see Supplementary Fig. 5). Note that the 1,500 nm pump photon has energy far below the 3.8–4.0 eV band gap of ITO[29] and hence produces an intraband excitation of the conduction band electrons. A white-light probe covering the range of 360–750 nm was employed and focused on the sample to a diameter of 396 μm in the short-delay-TA experiments and 220 μm in the long-delay-TA experiments. We observed a sub-picosecond transient response of $\Delta T(t)/T(0)$ in the short-delay-TA experiments, and a microsecond transient response in the long-delay-TA experiments. Here $\Delta T(t)$ is defined as $T(t) - T(0)$ with $T(t)$ being the transmission at delay time $t$ and $T(0)$ being the static transmission (both referenced to air). In what follows we denote the sub-picosecond $\Delta T(t)/T(0)$ component as the fast component, which we will attribute to a hot-electron-induced change of permittivity with a time scale determined by electron-phonon coupling. In contrast, the microsecond component, designated as the slow component, is assigned to a permittivity change due to thermal and elastic effects of the lattice. Note that besides being a broadband spectral response, the fast and slow components have characteristic relaxation times differing by six orders of magnitude, far longer than the decay time contrast observed in the noble metal counterparts; in the latter case typically the fast, electron-dominated component is a few picoseconds and the slow, lattice-dominated component is hundreds of picoseconds[7,30–32]. In the next two sections we discuss the fast and slow components separately.

**The sub-picosecond component**. The results of short-delay-TA experiments are summarized in Fig. 2. Figure 2a shows a

color-coded $\Delta T(t)/T(0)$ spectral map for wavelength from 360 to 750 nm and delay times up to 1.5 ps. Figure 2b–d shows the $\Delta T(t)/T(0)$, $T(t)$, and $\Delta T(t)$ transient spectra, which are the differential change in transmission, the transmission and the absolute change in transmission, respectively; here the chosen delay time (denoted as $t_{e,0}$ in Fig. 2a) is that when $\Delta T(t)/T(0)$ goes through a maximum. Note that the $\Delta T(t)/T(0)$ spectra exhibit a spectrally oscillating line-shape, which, as is evident from the $T(t)$ spectra, arises from a pump-induced redshift of the transmission spectrum. Qualitatively, a zero-crossing wavelength of the $\Delta T(t)/T(0)$ spectrum corresponds to a maximum or minimum in the $T(t)$ spectra. We note that the redshift of the transmission spectrum, that has large slopes due to the coexistence of multiple transmission minima, yields a broadband and remarkable change of absolute transmission (Fig. 2d) that reaches a maximal positive (negative) value beyond 25% ( − 20%). We attribute the redshift of the transmission spectrum to a positive change of the real part of the relative permittivity, $\Delta\varepsilon'(\omega)$, as a stronger dielectric contrast between the nanorod and free space can lead to a larger phase difference accumulation, which subsequently yields the out-of-phase conditions at longer wavelengths and thereby a redshift of the transmission spectrum.

To estimate the fluence and wavelength dependent $\Delta\varepsilon'(\omega)$ observed in our TA experiments, we carried out waveguide simulations, in which we arbitrarily introduced $\Delta\varepsilon'(\omega)$ ranging from 0 to 1 on top of the static permittivity, and calculate the wavelength dependent mode index $n_{eff}(\omega)$ as a function of $\Delta\varepsilon'(\omega)$. This allows for the calculation of the spectral locations of the five transmission minima associated with these values of $\Delta\varepsilon'(\omega)$ using equation (1). The dependence of $n_{eff}(\omega)$ on $\Delta\varepsilon'(\omega)$

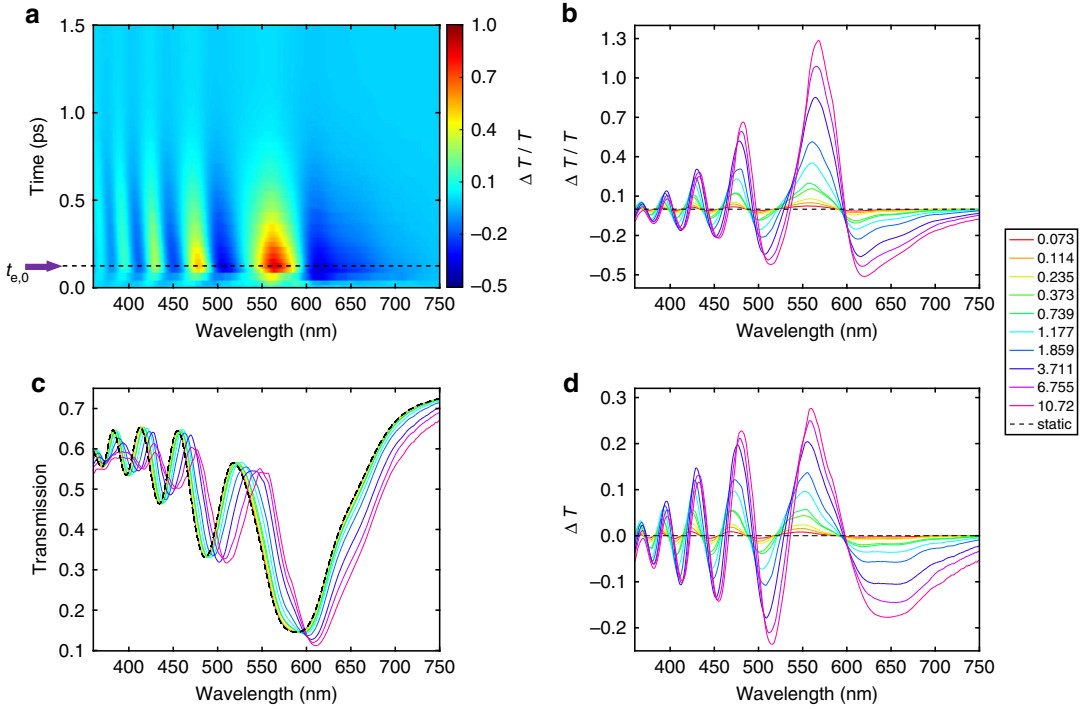

**Figure 2 | Sub-picosecond transient response dominated by the hot electrons.** (**a**) $\Delta T(t)/T(0)$ spectral map for the first 1.5 ps under a pump fluence of 3.71 mJ cm$^{-2}$. The purple arrow & black dashed line indicate $t_{e,0}$, which is the beginning of the fast component when the $\Delta T(t)/T(0)$ amplitude reaches a maximum. (**b–d**) Fluence dependent $\Delta T(t)/T(0)$, $T(t)$, and $\Delta T(t)$ spectra at $t_{e,0}$. The legend has a unit of mJ cm$^{-2}$ and applies to panel **b**, **c**, and **d**.

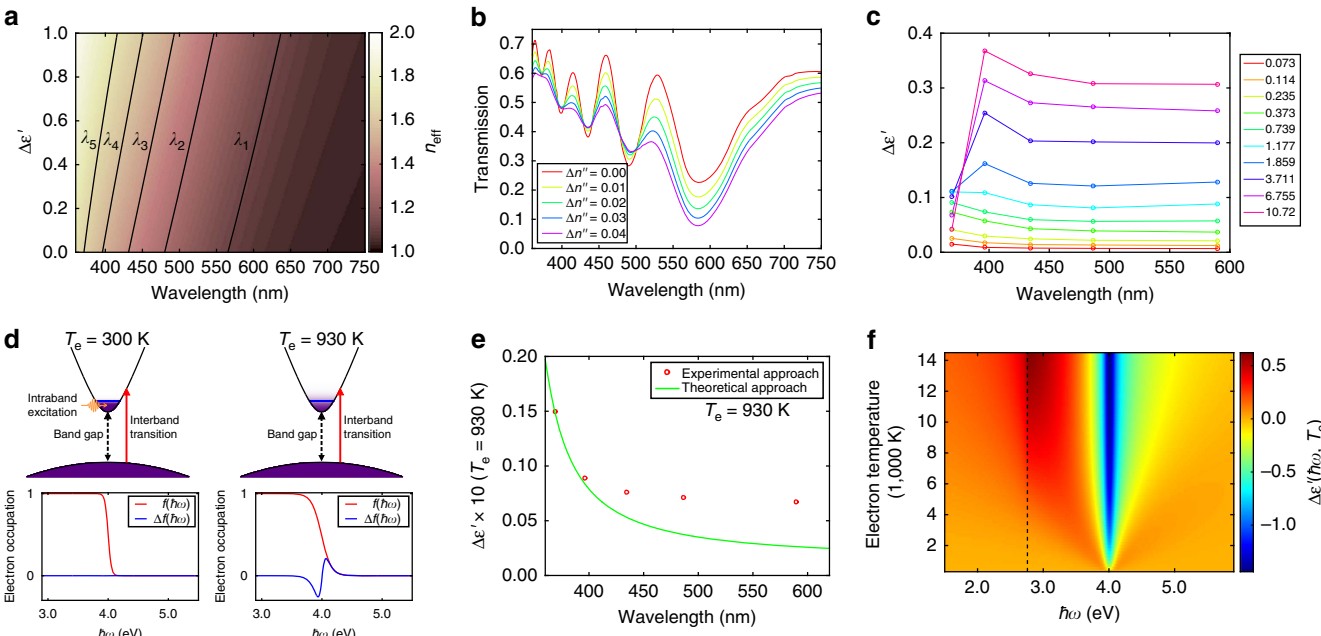

**Figure 3 | Optical nonlinearity in the visible range due to the hot electrons.** (**a**) Waveguide simulation predicted effective mode index as a function of wavelength and change of the real part of the relative permittivity (on top of the static value). Black solid lines indicate the calculated transmission minima wavelengths using equation (1). (**b**) Optical simulation predicted transmission spectra after adding various constant imaginary parts (independent of wavelength), $\Delta n''$, to the static refractive index of the ITO-NRA. (**c**) Fluence dependent $\Delta \varepsilon'(\omega)$ at $t_{e,0}$ (beginning of the fast component) obtained by the experimental approach. (**d**) Top: schematic band diagram showing the modification of interband transitions from 300 K (left) to 930 K (right). Bottom: The Fermi distribution and the change of Fermi distribution (with respect to the room temperature case) for the electron gas at 300 K (left) and 930 K (right). (**e**) Green curve shows the $\Delta \varepsilon'(\omega)$ obtained by the theoretical approach; red circles represent $\Delta \varepsilon'(\omega)$ obtained by the experimental approach. Both data are plotted for electron gas at 930 K, corresponding to the lowest pump fluence of 73 μJ cm$^{-2}$. (**f**) Photon energy ($x$ axis) and electron temperature ($y$ axis) dependent $\Delta \varepsilon'(\omega)$ obtained from the theoretical approach; the vertical dashed line indicates the optical transition associated with electrons excited from the valence band maximum to the conduction band minimum.

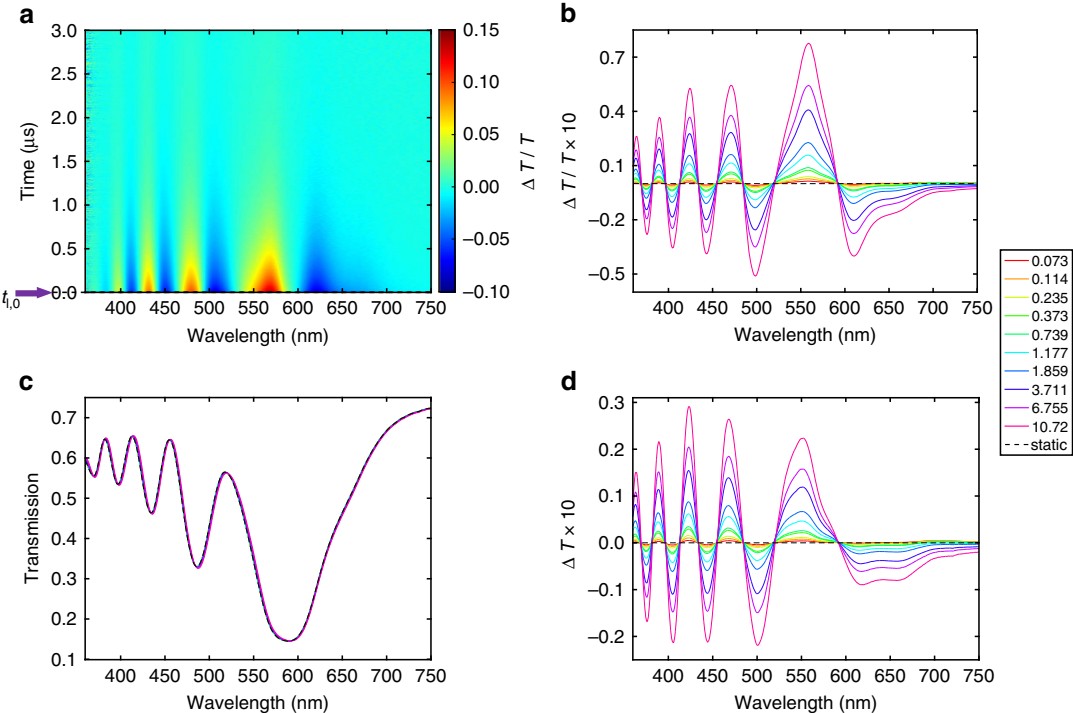

**Figure 4 | Microsecond transient response of the ITO-NRA due to the hot lattice.** (**a**) $\Delta T(t)/T(0)$ spectral map for the first 3 μs under a pump fluence of 26.3 mJ cm$^{-2}$. The purple arrow & black dashed line indicate $t_{l,0}$, which is the beginning of the slow component (corresponding to 850 ps delay time). (**b–d**) Fluence dependent $\Delta T(t)/T(0)$, $T(t)$, and $\Delta T(t)$ spectra at $t_{l,0}$. The legend has a unit of mJ cm$^{-2}$ and applies to panel **b**, **c**, and **d**.

and wavelength is color-coded in Fig. 3a, in which we also present the $\Delta\varepsilon'(\omega)$ dependent spectral locations of the transmission minima. We further confirmed by optical simulations that changing the imaginary part of the refractive index, $\Delta n''$ (or effectively the imaginary part of the permittivity), does not shift the transmission minima, as presented in Fig. 3b. As a result, by correlating the experimentally observed redshifts of the transmission minima shown in Fig. 2c with the $\Delta\varepsilon'(\omega)$ dependent shifts of the transmission minima shown in Fig. 3a, we can deduce the experimental $\Delta\varepsilon'(\omega)$ at the transmission minima wavelengths $\lambda_1$ to $\lambda_5$; these are plotted in Fig. 3c for the fast component. This approach of combining the experimentally observed redshifts of transmission minima ($\Delta\lambda_m$) and the waveguide simulation predicted relation of $\Delta\varepsilon'$ versus $\lambda_m$ to deduce the $\Delta\varepsilon'(\omega)$ in our TA experiments is denoted as the experimental approach; note that this approach does not provide quantitative information about $\Delta\varepsilon''(\omega)$, which is the change of the imaginary part of the permittivity of ITO.

Examination of Fig. 3c reveals that at low pump fluences, $\Delta\varepsilon'(\omega)$ peaks at $\lambda_5$, which is close to ITO's band gap, and stays approximately constant at longer wavelengths. At high pump fluences, however, $\Delta\varepsilon'(\omega)$ peaks at $\lambda_4$, and falls off quickly at shorter wavelengths. Here we theoretically calculate $\Delta\varepsilon'(\omega)$ by considering the modification of interband transitions of ITO through intraband excitation, as schematically illustrated in Fig. 3d. The heating of the conduction band electrons results in a change of its distribution, which can be modelled in details using the procedure described earlier[33]. A change in the Fermi distribution function of the electron gas gives rise to a change of the imaginary part of the permittivity, $\Delta\varepsilon''(\omega)$, through the change of interband transitions from the filled valence band to the partially occupied conduction band, which in turn produces a change of the real part of the permittivity, $\Delta\varepsilon'(\omega)$, dictated by the Kramers–Kronig relation. This is in essence consistent with the results reported for noble metal nanoparticles under intraband

optical excitation[34]. This approach, which gives quantitative information about both $\Delta\varepsilon'(\omega)$ and $\Delta\varepsilon''(\omega)$, is denoted as the theoretical approach; details of this approach are discussed in Supplementary Note 4, as well as Supplementary Figs 6, 7, 8 and 9. In Fig. 3e we plot the theoretical approach predicted $\Delta\varepsilon'(\omega)$ under the constant electric-dipole matrix element approximation, as well as the $\Delta\varepsilon'(\omega)$ obtained from the experimental approach (as shown in Fig. 3c), both associated with the lowest experimental pump fluence of 73 μJ cm$^{-2}$ (in the weak perturbative regime) in our short-delay-TA experiments. A value for the matrix element squared of $2.8 \times 10^{-49}$ J kg was chosen to match the two data sets in Fig. 3e at $\lambda_5$; a larger experimental $\Delta\varepsilon'(\omega)$ in comparison to the theoretical $\Delta\varepsilon'(\omega)$ at longer wavelengths results from a permittivity change contributed by a reduction of the intrinsic plasma frequency of ITO due to an increase of the electron temperature[33], which has minimal effect at short wavelengths and is not captured by the interband absorption model. Using this same constant matrix element to calculate $\Delta\varepsilon'(\omega)$ for the entire electron temperature range (up to ~14,000 K) associated with our TA experiments yields Fig. 3f; the fluence dependent electron temperature is shown in Supplementary Fig. 10 (also see discussion in Supplementary Note 5). The one order of magnitude higher electron temperature reached in ITO-NRA, compared with the gold nanorod counterpart pumped at a similar fluence[8], is due to a much smaller electron heat capacity, which further results from ITO's low electron concentration in comparison to gold. We note that the magnitude of the theoretical $\Delta\varepsilon'(\omega)$ is within a factor of ~1.5 of the experimental $\Delta\varepsilon'(\omega)$ shown in Fig. 3c. In addition, Fig. 3f shows that at high pump fluences, $\Delta\varepsilon'(\omega)$ peaks at the photon energy that corresponds to the optical transition with an electron excited from the valence band maximum to the conduction band minimum; this $\Delta\varepsilon'(\omega)$ line-shape is qualitatively consistent with behaviour of the experimental $\Delta\varepsilon'(\omega)$ shown in Fig. 3c. Furthermore, Supplementary Fig. 8c shows that for optical

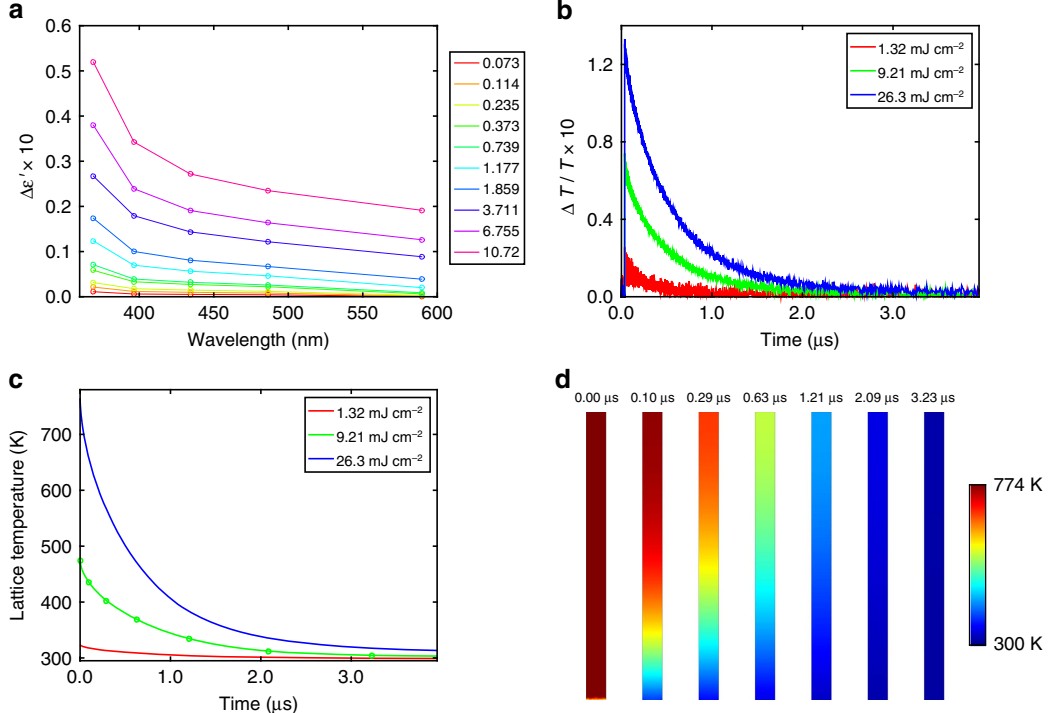

**Figure 5 | Permittivity change due to the hot lattice and temperature decay of the lattice.** (**a**) $\Delta\varepsilon'(\omega)$ at $t_{l,0}$ (beginning of the slow component) obtained by the experimental approach. Legend has a unit of mJ cm$^{-2}$. (**b**) Fluence dependent decay of $\Delta T(t)/T(0)$ at 560 nm plotted for delay times up to 4 μs measured in the long-delay-TA experiments. (**c**) Simulated decay of temperature averaged over the entire nanorod volume corresponding to the experimental pump fluences. (**d**) Snapshots of the temperature profiles of the nanorod (at the plane cutting through the centre of the nanorod) at different delay times indicated by the green circles in **c** under a pump fluence of 9.21 mJ cm$^{-2}$.

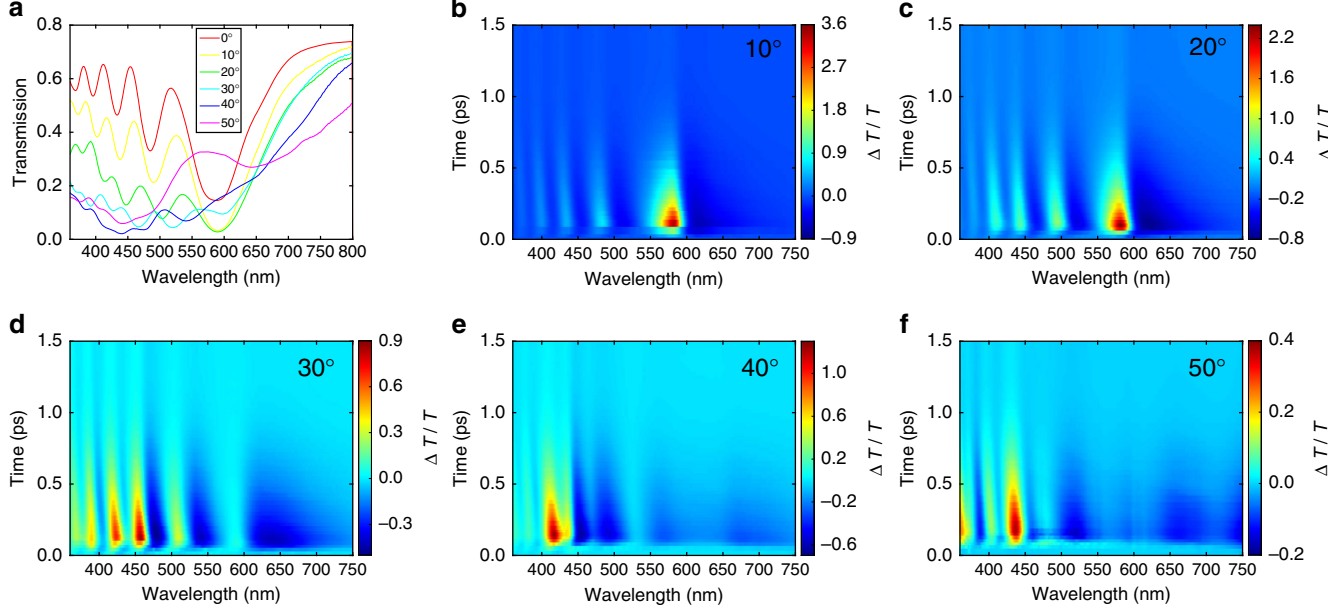

**Figure 6 | Dependence of the spectral features on the incidence angle.** (**a**) Static transmission spectra at various incidence angles (from 0° to 50° in 10° increments, referenced to air). (**b**–**f**) $\Delta T(t)/T(0)$ spectral maps measured at 10, 20, 30, 40 and 50°, plotted for delay times up to 1.5 ps. All measurements were performed for the (0, 0) order, and the pump fluence was $3.71 \times \cos\theta_{inc}$ mJ cm$^{-2}$, where $\theta_{inc}$ is the incidence angle.

transitions associated with excited electrons lying between the Fermi energy at room temperature and the conduction band minimum, $\Delta\varepsilon''(\omega)$ has positive values, indicating that rising of the electron temperature will cause a stronger optical absorption; this is in concert with Fig. 2c which shows diminished amplitudes

of transmission minima $\lambda_5$, $\lambda_4$, and $\lambda_3$ on the high energy side (Fig. 3b shows positive $\Delta\varepsilon''(\omega)$ leads to less pronounced transmission minima). Below the absorption threshold energy (which is the energy difference between the conduction band minimum and the valence band maximum), $\Delta\varepsilon''(\omega)$ stays zero

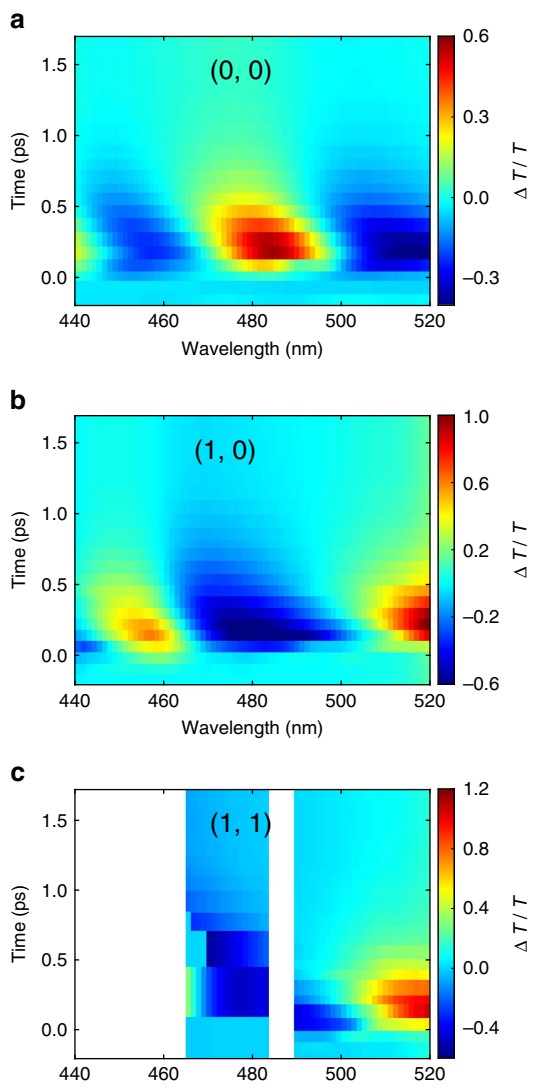

**Figure 7 | Redistribution of light intensity among different grating orders.** (**a**–**c**) $\Delta T(t)/T(0)$ spectral maps for the (0, 0), (1, 0) and (1, 1) grating orders, plotted for delay times up to 1.7 ps.

due to the existence of the band gap, and hence the optical absorption should be unchanged; this is consistent with the nearly unchanged amplitudes of the transmission minima $\lambda_1$ and $\lambda_2$ on the low energy side. We note that the disagreement between the experimental $\Delta\varepsilon'(\omega)$ shown in Fig. 3c and the theoretical $\Delta\varepsilon'(\omega)$ shown in Fig. 3f at high electron temperatures may arise from: (1) the implicit assumption that the entire ITO nanorod volume has a uniform electron temperature (as $\Delta\varepsilon'(\omega)$ was considered as a constant within the nanorod); (2) the non-ideality (and smearing) of the absorption around the absorption onset energy in comparison to the theoretical model (evident from Supplementary Fig. 8d), possibly arising from the broadening of states[29] and defect states; and (3) a breakdown of the constant matrix element approximation arising from the wide energy range considered in the theoretical model. However, further modelling was not attempted because the possible occupation of higher energy bands (above the considered conduction band) or other regions of the momentum space[35,36] by the hot electrons at very high temperatures. In fact, the modelling difficulty originates from the doping dependent dispersion diagram[35,36], and more importantly, a far lower electron concentration in ITO-NRA,

which leads to a much wider range of energies that are being accessed by the hot electrons under the pump fluences in our TA experiments; this is in sharp contrast to the noble metal case, where the assumption of a constant matrix element is more valid due to the lower electron temperatures reached[8,37,38].

**The microsecond component.** We now discuss the slow component of $\Delta T(t)/T(0)$. Figure 4a shows a $\Delta T(t)/T(0)$ spectral map acquired from the long-delay-TA experiments; note that now the delay time is up to 3 μs. The $\Delta T(t)/T(0)$, $T(t)$ and $\Delta T(t)$ spectra at $t_{l,0}$ (defined as 850 ps delay time) acquired from the short-delay-TA experiments, are shown in Fig. 4b–d, respectively. This 850-ps delay time can be taken as the beginning of the microsecond decay, when all the absorbed pump energy still resides in the lattice. The reason for presenting the transient spectra at 850 ps rather than those at earlier delay times after the fast component fully decays (in about a picosecond), is due to the excitation of coherent acoustic vibrations that give an additional contribution to the $\Delta T(t)/T(0)$ spectra during the first $\sim 800$ ps (which is approximately equal to the dephasing time of the coherent vibrations), as shown in Supplementary Fig. 11. Also, note that the transient spectra at 850 ps were acquired from the short-delay-TA experiments, as the long-delay-TA experiments have poorer temporal resolution ($\sim 100$ ps) and lower signal to noise ratio than the short-delay-TA experiments. The amplitude of $\Delta T(t)/T(0)$ for the slow component shown in Fig. 4b is about one order of magnitude smaller than the fast component analogue presented in Fig. 2b. This is correlated with a much smaller shift of the transmission spectra and absolute change of transmission which are shown in Fig. 4c,d, respectively. The fluence dependent $\Delta\varepsilon'(\omega)$ at $t_{l,0}$ shown in Fig. 5a, which is obtained using the experimental approach (as was used for estimating $\Delta\varepsilon'(\omega)$ for the fast component shown in Fig. 3c), is found to peak at the shortest wavelength and fall off quickly at longer wavelengths for all pump fluences. The fall-off of $\Delta\varepsilon'(\omega)$ at long wavelength arises because at $t_{l,0}$ the electron gas is in thermal equilibrium with the lattice; hence the reduction of the plasma frequency caused by the hot electron gas is negligible (as is evident from the small lattice temperature rise in comparison to the large electron temperature rise shown in Supplementary Fig. 10). In fact comparison of Fig. 2b and Fig. 4b reveals that the amplitude of $\Delta T(t)/T(0)$ at $t_{e,0}$ has a larger spectral weight at longer wavelengths compared with that at $t_{l,0}$ (this is also observed in measurements on other ITO-NRA samples, as shown in Fig. 8). The peak-at-shortest-wavelength at $t_{l,0}$ is likely due to a small electron temperature rise, therefore electron occupation far from the Fermi energy is largely unchanged and $\Delta\varepsilon''(\omega)$ is expected to be confined to a narrow spectral range around the Fermi energy (as shown in Supplementary Fig. 8c at the low electron temperature range); correspondingly, $\Delta\varepsilon'(\omega)$ peaks at higher energy, similar to the behaviour of $\Delta\varepsilon'(\omega)$ observed at lower pump fluences for the fast component. We note that various thermal effects, including a possible decrease of band gap with an increased lattice temperature[39], together with elastic response (such as thermal expansion) of ITO can contribute to $\Delta\varepsilon'(\omega)$ of the slow component; therefore further theoretical modelling was not attempted due to the poorly understood temperature dependent band structure and strain dependence of the permittivity that is outside the scope of the current study.

The microsecond decay time of the slow component (best illustrated by Fig. 5b that shows the $\Delta T(t)/T(0)$ kinetics at 560 nm) is far slower than the few-hundred-picosecond decay time observed in solution-based or substrate-supported plasmonic structures of noble metals[40,41]. This long decay time can be attributed to a lower thermal conductivity of ITO[42] in

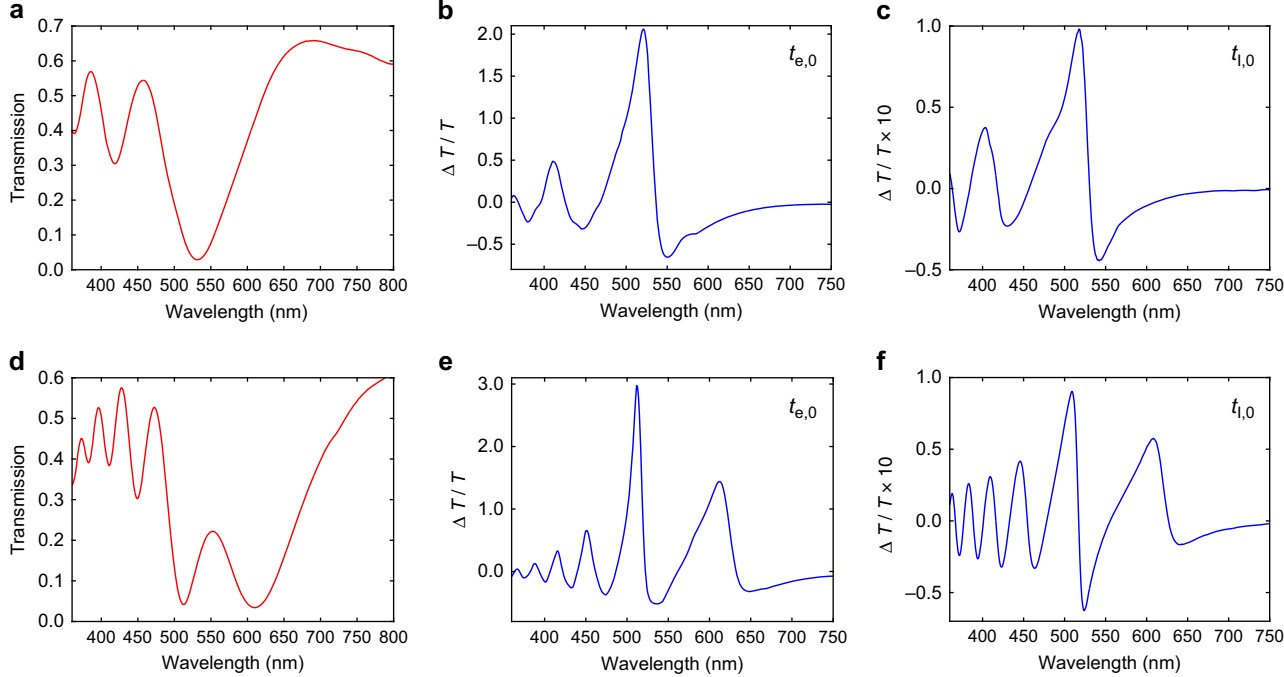

**Figure 8 | Tuning the spectral features by adjusting the height of the nanorod.** Static transmission spectrum, $\Delta T(t)/T(0)$ spectrum at delay time $t_{e,0}$, and $\Delta T(t)/T(0)$ spectrum at delay time $t_{l,0}$ (850 ps). (**a–c**) are for an ITO-NRA sample with 1.4 μm height. (**d–f**) are for an ITO-NRA sample with 2.9 μm height. Pump fluence was 7.4 mJ cm$^{-2}$.

comparison to noble metals (Supplementary Note 5 and Supplementary Fig. 10), and a small contact area with the surrounding medium (YSZ substrate in the present study) for heat dissipation (owing to the large aspect-ratio). To get a quantitative understanding of the slow decay we performed heat-transfer simulations, in which we assumed a uniform, initial lattice temperature rise of the ITO nanorod. This assumption is valid because the Fermi velocity of the electron gas in ITO is initially greater than its room temperature value on the order of ~1 × 10$^6$ m·s$^{-1}$ (estimated from $[\mu(300\,K)/m]^{1/2}$ with $m = 0.263\,m_e$ (ref. 43) and $\mu(300\,K) \approx 1.2$ eV where $m_e$ is the free electron mass); therefore during the first 1 ps (fast component) the electron gas can travel a distance on the order of the length of the nanorods. Details of the lattice temperature rise calculation and heat-transfer simulation can be found in Supplementary Notes 5 and 6, respectively (also see Supplementary Fig. 12). Figure 5c presents the simulated average lattice temperature versus the decay time; note that the temporal characteristics are in good quantitative agreement with the experimental $\Delta T(t)/T(0)$ decay shown in Fig. 5b. In Fig. 5d we further present the simulated temperature profiles of the nanorod at different delay times. We note that soon after the initial response, the temperature appears to be monotonically decreasing from the top to the bottom of the nanorod, indicating that the length of the nanorod acts as a bottleneck for heat dissipation. It is expected that the kinetics of the lattice temperature induced changes in $\Delta T(t)/T(0)$ can potentially be used to indirectly deduce the thermal conductivity of other uniform one-dimensional nanostructures[44].

**Spectral tunability and beam-steering capability.** Figure 6 summarizes the static and transient spectral response of the ITO-NRA under different incidence angles. Notably, Fig. 6a shows that increasing the incidence angle leads to a redshift of the static transmission spectrum, and at high angles the transmission

minima become less pronounced. This angular dependent spectral feature can be qualitatively explained by a modified version of equation (1), $\lambda_m \cdot \cos\theta_{inc} = 2h(n_{eff} - 1)/(2m - 1)$ where $\theta_{inc}$ is the incidence angle. Here the effective wavelength of light along the length of the nanorod is altered by a factor of $\cos\theta_{inc}$, hence the transmission minima at oblique incidence angles are expected to occur at longer wavelengths compared with the normal incidence case. This static trend is consistent with the angular dependent $\Delta T(t)/T(0)$ spectral maps (shown in Fig. 6b–f); specifically, under a given incidence angle the spectral line-shape and amplitude of each $\Delta T(t)/T(0)$ spectral map are dictated by the redshift of the corresponding static transmission spectrum.

In addition to the measurements on the (0, 0) order, we performed additional short-delay-TA experiments to analyse one of the four equivalent (1, 0) and (1, 1) diffraction spots (See Supplementary Fig. 1a). Examination of the spectral maps for the higher diffraction orders (Fig. 7b,c) and that of the zero order (Fig. 7a) reveals that the higher-order $\Delta T(t)/T(0)$ signals have opposite signs in comparison to the zero order. This indicates that the ITO-NRA dynamically redistributes the light intensity among different grating orders following the pump excitation, and further suggests that ITO (and possibly other TCO materials) can be utilized in the design of active optical components, such as a dielectric metasurface[45,46], to realize the control of intensity, phase and polarization, and with it the holographic response of light in the visible range.

To further demonstrate the spectral tunability achievable by adjusting the geometric parameters, we performed static and short-delay-TA experiments on two additional ITO-NRAs with nanorod heights of 1.4 and 2.9 μm, respectively (the SEM images, near-infrared transmission spectra and visible $\Delta T(t)/T(0)$ spectral maps for these two ITO-NRA samples are presented in Supplementary Fig. 13). The static transmission spectra of these two samples, depicted in Fig. 8a,d, show that increasing the nanorod height can produce a larger number of transmission minima and redshift the first transmission minimum wavelength

                                                                                                     

($\lambda_1$); these are consistent with equation (1) that relates the wavelengths of transmission minima to the height of the nanorod. The transient spectra at $t_{e,0}$ (shown in Fig. 8b,e) and $t_{l,0}$ (shown in Fig. 8c,f) for these two samples again arise from redshifts of the corresponding static transmission spectrum, consistent with the earlier discussions.

## Discussion

In conclusion, we have shown that the optical nonlinear response of ITO-NRAs enables all-optical modulation over a wide spectral range from 360 to 710 nm with large absolute amplitude at both the sub-picosecond and microsecond time scales. The sub-picosecond modulation, which is an order of magnitude faster than the noble metal systems, arises from the unique electronic structure of ITO which permits an efficient resonant near-infrared pumping that modulates its permittivity in the visible (dielectric) range. We theoretically modelled the electron temperature dependent permittivity change, obtaining semi-quantitative agreement with the permittivity change obtained by comparing the experimental and simulation results. On the other hand, the origin of the microsecond modulation, which is three orders of magnitude slower than the noble metal counterparts, is ascribed to the low thermal conductivity and large aspect-ratio of the ITO-NRAs. We further demonstrated that the all-optical modulation can be spectrally tuned, simply by changing the angle of incidence on the array or tailoring the height of the nanorods. In addition, the nonlinear response can allow for a dynamic redistribution of light intensities along different directions associated with the grating orders. Further improvements of the modulation intensity may be achievable by specially designed high quality-factor optical resonators[47] operating at a narrower bandwidth, adjusting the chemical compositions and doping concentrations[48], accessing materials' optical response at the ε-near-zero-regime[49,50], or exploiting other pumping schemes such as interband pumping of electrons from the valence to the conduction band[23,51]. The observed permittivity modulation in the transparent spectral region obtained by pumping in the metallic range can be expected to lead to the development of novel active optical components.

## Methods

**Sample fabrication.** Briefly, an epitaxial ITO film of 10 nm thickness was deposited on YSZ (001) substrate using magnetron sputtering at 600 °C, 5 mTorr under 20 sccm Ar gas flow. A 70 nm thick GL-2000 electron beam resist (Gluon Labs) was then spin coated on the substrate, followed by exposure of an array of 150 nm dots with designed pitch sizes (JEOL JBX-9300FS electron beam lithography system). The exposed sample was developed in Xylenes at room temperature for 60 s, and then rinsed by IPA. 2 nm Cr and 15 nm Au was thermally evaporated on the sample, which was subsequently lifted off in Anisole at 75 °C for 1 h. The nanorod growth was performed at a customized tube furnace system[33].

**Steady state measurements.** Transmission spectra in the near-infrared range were measured with FTIR (Thermo Nicolet 6700). A pair of ZnSe lenses were used to focus the light down to a 1-mm-diameter spot. Transmission spectra in the visible range were measured with an ultraviolet/vis/near-infrared spectrophotometer (Perkin-Elmer Lambda 1050).

**Transient absorption measurements.** Transient absorption experiments with delay times up to 1,000 ps were performed using a 35 fs amplified titanium:sapphire laser operating at 800 nm at a 2 kHz repetition rate. Pump pulses at 1,500 nm were generated via a white light seeded optical parametric amplifier and were reduced in repetition rate to 1 kHz. Broadband probe pulses were generated by focusing a portion of the amplifier output into a CaF₂ window (2 mm thick). The probe pulses were mechanically time-delayed using a translation stage and retroreflector. The pump spot diameter on the sample was 396 μm. Full spectral maps for the (0 ,0) order appear in Supplementary Fig. 14 and Supplementary Note 7. Representative $\Delta OD(t)$ and $T(t)$ spectral maps appear in Supplementary Fig. 15 and Supplementary Note 8. An optical fibre was used for measurements of the (1, 0) and (1, 1) diffraction orders. Due to the large spatial footprints, only narrow spectral windows, shown in Fig. 7b,c, were collected for these higher orders.

Longer time-delay transient absorption measurements were performed with ~100 ps time resolution using a 100 fs pump pulse and an electronically delayed white light probe pulse. The probe pulse is generated via self-phase modulation of a Nd:YAG laser in a photonic crystal fibre. Instabilities in the probe pulse were compensated by monitoring a beam-split portion of the pulse in a separate detector. Signal to noise ratios achieved with this system are notably lower than that those obtained for the higher time-resolution transient absorption system, primarily owing to the lower probe pulse-to-pulse stability. The pump spot diameter on the sample is 220 μm. Full spectral maps for the (0 ,0) order appear in Supplementary Fig. 16 and Supplementary Note 7.

**Finite-element simulations.** The optical simulation and waveguide simulation were performed with the Wave Optics module of COMSOL Multiphysics. The optical simulation was full three-dimensional simulation in which periodic boundary conditions were applied along the in-plane directions; transmission and reflection of the ITO-NRA can be obtained. The waveguide simulation was a two-dimensional simulation, in which a eigenmode analysis was performed on the cross-section of ITO nanorod for calculating the effective mode index. The heat-transfer simulation was enabled by the heat-transfer module of COMSOL Multiphysics. More details about optical, waveguide and heat-transfer simulations appear in Supplementary Notes 1, 2 and 6.

**Data availability.** All relevant data used in the article are available from the authors.

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

## Acknowledgements

The work was funded by the MRSEC program (NSF DMR-1121262) at Northwestern University. Use of the Center for Nanoscale Materials was supported by the U.S. Department of Energy, Office of Science, Office of Basic Energy Sciences, under contract no. DE-AC02-06CH11357. This work made use of the EPIC facility of the NUANCE Center at Northwestern University, which has received support from the Soft and Hybrid Nanotechnology Experimental (SHyNE) Resource (NSF NNCI-1542205); the MRSEC program (NSF DMR-1121262) at the Materials Research Center; the International Institute for Nanotechnology (IIN); the Keck Foundation; and the State of Illinois, through the IIN. We thank insightful discussions with Prof. Hui Fang, Dr Shi-Qiang Li, Dr Ankun Yang and Mr. Guohua Wei.

## Author contributions

P.G., R.D.S. and R.P.H.C. designed research. P.G. fabricated the samples, performed static measurements and simulations, developed the theoretical models and performed the calculations. R.D.S. and B.T.D. performed the TA experiments. J.B.K. contributed to the analysis. L.E.O. assisted the electron beam lithography experiments. P.G. wrote the manuscript with input and editing from all authors. R.P.H.C. and R.D.S. supervised the project.

## Additional information

**Competing financial interests:** The authors declare no competing financial interests.

**How to cite this article**: Guo, P. *et al.* Large optical nonlinearity of ITO nanorods for sub-picosecond all-optical modulation of the full-visible spectrum. *Nat. Commun.* **7:**12892 doi: 10.1038/ncomms12892 (2016).

