## [Peer review file · Nature Communications]

Reviewers' comments:

Reviewer #1 (Remarks to the Author):

This manuscript reports on the ultrafast transient response of periodic arrays of standing ITO nanorods. The dynamics of the array is induced and characterized through its resonant response governed by both a localized surface plasmon mode and several Fabry-Perot cavity modes. The arrays show two characteristic relaxation regimes spanning 6 orders of magnitude, from the ps to the us timescale, following pumping at the LSPR supported by individual nanorods. The fast ps transient is attributed to electron scattering and assigned to changes in the permittivity (the real part mainly) of the ITO nanorods. The slower dynamics is the result of phonon scattering. The contrasted behaviour compared to noble metal based plasmonic systems, which dynamics is within the ps timescale, is mainly attributed to the relatively large thermal conductivity of ITO. Another interesting finding is in the possibility of dynamically modulating the power diffracted by the arrays in different orders, although this aspect of the work was not developed to the full in the present report, mostly emphasizing the fundamental (at the nanostructured material level) rationalization of the experimental observations.

I have found the manuscript of excellent quality, especially the experimental results presented in fig2-3 are quite stunning. The analysis is robust and both numerical modelling and theoretical background, as well as additional experimental work, are extensively detailed in the SI, which I found very instructive. Overall, I think the results reported here, along with their discussion, are of interest to the Nature Comm. readership and should be warranted publication following the response from the Authors to the following minor comments:

1) How would the slow dynamics (we know the amplitude of the effect is small in the first place) be impacted by tuning the pump wavelength off the LSPR and into one of the Fabry-Perot resonances, or even just simply off resonance?

2) I think the claimed link between the 'waveguided' mode and the standing wave resonances observed in fig. 1b etc. should be made unambiguous from the start (page 4, second paragraph) for example by simply stating that the minima/maxima observed in the optical properties are standing wave resonances based on the HE₁₁ waveguided mode. On the same note, since the Authors discuss the effective index of the mode, it may be more appropriate to replace β/β_0 by n_{eff} . Also, I am unsure about the value of the space-averaged field distributions plotted in S5. Why take a spatial average?

3) The reflection spectrum $R(0,0)$ in figure 1 should have its own axis, for example as a secondary axis on the right-hand side, or the y-axis label "transmission" should be removed. In any case, something needs to be done in this figure to convey a clear message, as it is confusing in the present form.

The same remark holds for plots in the SI (S13), showing transmission, reflection, and absorption, but labelling transmission only.

4) It is awkward to discuss fig. 4-5 before fig. 3.

5) Please indicate the content of the legend in fig. 4. : the imaginary part of the refractive index k should be explicitly named both in the main text and in the figure caption.

6) In the SI, it would have been useful to show the material dispersion of ITO throughout the spectral range covered in the study (visible + IR) and not only through the visible range. In particular I think it is necessary to show the frequency at which ITO becomes metallic and for which real part of the permittivity the rods support the LSPR, as claimed in the text. The same remark can be made for fig. 5c.

Reviewer #2 (Remarks to the Author):

A. Summary of the key results

The authors report on "Large Optical Nonlinearities of Indium-Tin-Oxide Nanorods for Sub-Picosecond All-Optical Modulation of the Full Visible Spectrum".

The optical nonlinearities of nanostructures were induced by intense optical excitation of mobile electrons and provide dynamic tuning of their electromagnetic responses. This allows for optically controlled ultrafast signal switching and processing on a sub-picosecond time scale. The source of the required nonlinearities (of indium-tin-oxide nanorod arrays (ITO-NRAs)) is intraband, on-plasmon-resonance optical pumping, which enables all-optical modulation of the full visible range (from 360 nm to 710 nm). The results seem to demonstrate that all-optical control of the visible spectrum can be achieved with degeneratively doped wide-bandgap semiconductors in their transparent regime with speeds faster than typical noble metal counterparts.

B. Originality and interest: if not novel, please give references

The material is novel and of interest.

C. Data & methodology: validity of approach, quality of data, quality of presentation

Synthesis of ITO Arrays, experimental data and simulations are well done and with a sufficient quality of presentation.

D. Appropriate use of statistics and treatment of uncertainties

Not really applicable here.

E. Conclusions: robustness, validity, reliability

The performed experiments together with the former works of the author and cited literature suggest that the data is sufficiently robust and repeatable.

F. Suggested improvements: experiments, data for possible revision

- The transmission data in figure 1 b and c are non overlapping (360 -710 nm and 800 - 2400 nm). Even in the light that there is not much happening in between, the gap is annoying.

- While partially done throughout the text, the authors should painstakingly distinguish between single particle and array effects.

The "Spectral tunability and beam-steering capability" subchapter is disappointing and short. The authors would improve their manuscript if this section would be rewritten and results better emphasized.

G. References: appropriate credit to previous work?

Yes

H. Clarity and context: lucidity of abstract/summary, appropriateness of abstract, introduction and conclusions

My only concern is the following: The authors use the phrase "thereby allowing for ultrafast signal switching and processing by optical means" in their abstract.

However, neither (sub)-picosecond (order of one ps) is really ultrafast from the optical perspective, nor have the authors really demonstrated information processing. A less overemphasizing way would be to clearly state the two different timescales involved sub (ps and μ s) and to write that the demonstrated properties might be utilized in the future for information processing.

Reviewer #3 (Remarks to the Author):

I reviewed the optical study of ITO nano-structure work done by Prof. Robert P. H. Chang group. The current paper carefully studied the optical transmittance of nano-structured ITO at visible regime. They performed both fast and slow transient studies and developed theoretical modelling for the experimental explanations. They clearly demonstrated the capability to achieve transmittance signal modulation at visible frequencies up to 20% using the near-IR pumping induced plasmonic resonance effects. The data analysis is clear, the measurement results are convincing and their study is systematic and complete. I think this is really a good work and thus I recommend to publish after a revision.

Below there are several questions and comments I wish to ask:

It is known that metals are not an ideal platform for subwavelength light manipulation, and this becomes even obvious in the ultrafast time scale switching. The current developed metallic nanostructure certainly represents one strategy to overcome these issues. I also noticed that very recently, the authors had studied ultrafast tunability at mid-IR and near-IR regime (*Nature Photonics*, 10, 267 (2016)). Thus it is good to see a full picture over wide frequencies of this system. In the meantime, there also has other development using novel functional material, such as graphene and metamaterials for example (*Nature Photonics*, 10, 244 (2016); *Appl. Phys. Lett.* 104, 141102 (2014)). It would be great if the authors could cite and comment on these developments along this direction.

The current reported maximum light manipulation in the visible regime is still less than 30%, can the authors comment on what are the limiting factors, this would be helpful for the general readers and possibly new works following the current study.

Below are some of my detailed comments and suggestions to change:

- 1) One important aspect need to improve is that the transition from ΔT to $\Delta \epsilon$ is a bit abrupt, and it is not entirely clear how does the $\Delta \epsilon$ (both real and imag) are deduced and calculated. A detailed explanations and formula are recommended.
- 2) The authors mentioned from time to time the finite element simulations, but I missed the details explanation for this part, it would be good to add this;
- 3) The hot electron temperature is estimated to be as high as 14000 K, how does this temperature compared with others, can the authors comment on why it is so high?
- 4) How does ITO NRA doping concentration would play a role in the current studies?
- 5) Eq.1 might be confusing to the readers. Both the effective index and transmission minima are all denoted by n .
- 6) There are some typos in the text files in the SI;

Reviewer #1 (Remarks to the Author):

This manuscript reports on the ultrafast transient response of periodic arrays of standing ITO nanorods. The dynamics of the array is induced and characterized through its resonant response governed by both a localized surface plasmon mode and several Fabry-Perot cavity modes. The arrays show two characteristic relaxation regimes spanning 6 orders or magnitude, from the ps to the us timescale, following pumping at the LSPR supported by individual nanorods. The fast ps transient is attributed to electron scattering and assigned to changes in the permittivity (the real part mainly) of the ITO nanorods. The slower dynamics is the result of phonon scattering. The contrasted behaviour compared to noble metal based plasmonic systems, which dynamics is within the ps timescale, is mainly attributed to the relatively large thermal conductivity of ITO. Another interesting finding is in the possibility of dynamically modulating the power diffracted by the arrays in different orders, although this aspect of the work was not developed to the full in the present report, mostly emphasizing the fundamental (at the nanostructured material level) rationalization of the experimental observations.

I have found the manuscript of excellent quality, especially the experimental results presented in fig2-3 are quite stunning. The analysis is robust and both numerical modelling and theoretical background, as well as additional experimental work, are extensively detailed in the SI, which I found very instructive. Overall, I think the results reported here, along with their discussion, are of interest to the Nature Comm. readership and should be warranted publication following the response from the Authors to the following minor comments:

We thank the Reviewer for his/her keen interest in our work and address the comments in full below.

1) How would the slow dynamics (we know the amplitude of the effect is small in the first place) be impacted by tuning the pump wavelength off the LSPR and into one of the Fabry-Perot resonances, or even just simply off resonance?

The sequence of the dynamics is: 1) The conduction band electrons are heated by the near-infrared, on-resonance pump; 2) The electrons transfer their energy to the lattice, causing a lattice temperature rise; 3) The lattice and electrons are in thermal equilibrium, and gradually cool down to room temperature. Note that step 3) is associated with the slow dynamics. As a result, the effect of pump wavelength selection on the slow dynamics is dictated by how the pump excites the conduction band electrons in the first place. In our TA experiments, the on-plasmon-resonance pumping yields strong absorption of pump energy by the conduction band electrons and with it a significant electron temperature rise. Tuning the pump wavelength to one of the Fabry-Perot resonance or simply off resonance is expected to result in a much weaker pump absorption by the electrons (as we have shown in the manuscript that at the Fabry-Perot resonances the sample do not strongly absorb), and thereby a small lattice temperature rise. In the revised manuscript, we added the transient response of the sample at 568 nm (the wavelength at which the $\Delta T/T$ is maximized), when the sample is pumped at 800 nm; the results are shown as Supplementary Fig. 16; it is noteworthy that the

signal of the fast component is notably much smaller than the on-resonance pumping case. To clearly illustrate this point, we also added the sentence “Pumping the sample at 800 nm (off-resonance) was found to give significantly weaker response in comparison to the on-resonance pumping (Supplementary Fig. 16)” to the “Transient absorption experiments of the ITO-NRAs” paragraph in the revised manuscript.

2) I think the claimed link between the 'waveguided' mode and the standing wave resonances observed in fig. 1b etc. should be made unambiguous from the start (page 4, second paragraph) for example by simply stating that the minima/maxima observed in the optical properties are standing wave resonances based on the HE₁₁ waveguided mode.

We agree that this will improve the clarity of the discussion of the visible resonances. We have combined the two paragraphs in the original manuscript into a single paragraph in the revised manuscript, appearing after the subheading “static spectral features of the ITO-NRA”, and changed the sentence “arise from destructive interference between the waves propagating in the nanorod and the surrounding free space at different phase velocities” to “are simply standing wave resonances based on the HE₁₁ waveguide mode”.

On the same note, since the Authors discuss the effective index of the mode, it may be more appropriate to replace β/β_0 by n_{eff} .

The suggestion on replacing the beta terms by the effective mode index is excellent. In fact, this way we do not need to introduce the beta terms at all, hence the clarity can be improved. As such, we have now removed the use of the beta terms, and used n_{eff} (effective mode index) in the equations as well as discussions throughout the main text and the Supplementary Info.

Also, I am unsure about the value of the space-averaged field distributions plotted in S5. Why take a spatial average?

Because as shown in Fig. 1d, at the five transmission minima wavelengths the waves propagate at different velocities, it is not possible to choose a single cross-sectional plane perpendicular to the long axis of the nanorod, in which the electric field is at a maximum for all wavelengths. Therefore, we took the spatial average of the electric field, which can clearly demonstrate the pattern of the mode at these wavelengths, while avoiding potential confusions.

3) The reflection spectrum $R(0,0)$ in figure 1 should have its own axis, for example as a secondary axis on the right-hand side, or the y-axis label "transmission" should be removed. In any case, something needs to be done in this figure to convey a clear message, as it is confusing in the present form. The same remark holds for plots in the SI (S13), showing transmission, reflection, and absorption, but labelling transmission only.

This is an excellent observation. We have now changed the vertical axis to “Intensity” for all figures that include not only transmission but also reflection and/or absorption curves (specifically, for Fig. 1f and Supplementary Fig. S10).

4) It is awkward to discuss fig. 4-5 before fig. 3.

In the revised manuscript, we have moved the entire discussion of the slow component after the discussion of the fast component, and rearranged the panels of Fig. 3, Fig. 4 and Fig. 5; now Fig. 2 and Fig. 3 are dedicated to the fast component, whereas Fig. 4 and Fig. 5 are dedicated to the slow component.

5) Please indicate the content of the legend in fig. 4.: the imaginary part of the refractive index k should be explicitly named both in the main text and in the figure caption.

We have now explicitly defined the imaginary part of the refractive index as n'' ; this is also consistent with the theoretical derivation in the Supplementary Info. In the legend of Fig. 4, the change of the imaginary part of the refractive index is now denoted as $\Delta n''$.

6) In the SI, it would have been useful to show the material dispersion of ITO throughout the spectral range covered in the study (visible + IR) and not only through the visible range. In particular I think it is necessary to show the frequency at which ITO becomes metallic and for which real part of the permittivity the rods support the LSPR, as claimed in the text. The same remark can be made for fig. 5c.

In our manuscript, we used a Drude-Lorentz model to simply obtain reasonable wavelength dependent permittivity in the visible range for the mode index calculations, which is the basis for estimating the permittivity modulation. However, we did not expect the specific formula we used can be extended to a wider spectral range into the ultraviolet or near-infrared. In fact, a best fit of the near-infrared transmission spectrum was obtained in our earlier manuscript (Nat. Photonics, 2016, 10, 267-273), in which we used a Drude model. The need to use two models arises from the wide spectral range considered in our studies. Also note that the Drude-formula yielded static plasma frequency was used in our present study for calculating the electron distribution as a function of temperature.

We think it is a good suggestion to add the permittivity in the near-infrared range and show the zero-crossing wavelength of the real permittivity, as our work emphasizes access to both the dielectric and metallic regimes of ITO. Therefore, we have now replaced the refractive index plots for the visible range, which originally appeared in Supplementary Fig. S1, with permittivity plots in the near-infrared range appearing as Fig. S1c and S1d. Note that for consistency, we used the Drude model to generate the near-infrared permittivity (this is noted in the Figure legend), and the Drude-Lorentz model to produce the visible permittivity; there can be a slight discontinuity between visible and near-infrared permittivities.

Reviewer #2 (Remarks to the Author):

A. Summary of the key results

The authors report on "Large Optical Nonlinearities of Indium-Tin-Oxide Nanorods for Sub-Picosecond All-Optical Modulation of the Full Visible Spectrum". The optical nonlinearities of nanostructures were induced by intense optical excitation of mobile electrons and provide dynamic tuning of their electromagnetic responses. This allows for optically controlled ultrafast signal switching and processing on a sub-picosecond time scale. The source of the required nonlinearities (of indium-tin-oxide nanorod arrays (ITO-NRAs)) is intraband, on-plasmon-resonance optical pumping, which enables all-optical modulation of the full visible range (from 360 nm to 710 nm). The results seem to demonstrate that all-optical control of the visible spectrum can be achieved with degeneratively doped wide-bandgap semiconductors in their transparent regime with speeds faster than typical noble metal counterparts.

B. Originality and interest: if not novel, please give references

The material is novel and of interest.

C. Data & methodology: validity of approach, quality of data, quality of presentation

Synthesis of ITO Arrays, experimental data and simulations are well done and with a sufficient quality of presentation.

D. Appropriate use of statistics and treatment of uncertainties

Not really applicable here.

E. Conclusions: robustness, validity, reliability

The performed experiments together with the former works of the author and cited literature suggest that the data is sufficiently robust and repeatable.

F. Suggested improvements: experiments, data for possible revision

- The transmission data in figure 1 b and c are non-overlapping (360 -710 nm and 800 - 2400 nm). Even in the light that there is not much happening in between, the gap is annoying.

We have now added the transmission data between 710 nm to 800 nm in Fig. 1b. This is also done for the additional two ITO-NRA samples shown in Fig. 8a and 8d. Furthermore, we have expanded the wavelength range to 750 nm for all the other spectra shown in the manuscript (750 nm is the edge of spectral window we have captured in our TA experiments). However, it is noteworthy that from 710 nm to 750 nm the transmission modulation is small, as this region is spectrally far from the reddest transmission minimum, therefore we still state that all-optical modulation is obtained for the range from 360 nm to 710 nm.

- While partially done throughout the text, the authors should painstakingly distinguish between single particle and array effects.

In the revised manuscript, we have added the following sentences, “While the NIR LSPR is a localized phenomenon, the transmission minima in the visible range are due to coherent light diffraction by the ITO-NRA and therefore is attributed to an array effect”, to the beginning of the second paragraph of the section “Static spectral features of the ITO-NRA”.

The "Spectral tunability and beam-steering capability" subchapter is disappointing and short. The authors would improve their manuscript if this section would be rewritten and results better emphasized.

We thank for the excellent suggestion. In fact, these two properties exhibited by ITO-NRA are unique aspects of our work. In our revised manuscript, we rewrote the section “Spectral tunability and beam-steering capability” by adding Fig. 6, Fig. 7 and Fig. 8 (those originally appeared in the Supplementary Info) along with discussions of the figures.

G. References: appropriate credit to previous work?

Yes

H. Clarity and context: lucidity of abstract/summary, appropriateness of abstract, introduction and conclusions

My only concern is the following: The authors use the phrase "thereby allowing for ultrafast signal switching and processing by optical means" in their abstract. However, neither (sub)-picosecond (order of one ps) is really ultrafast from the optical perspective, nor have the authors really demonstrated information processing. A less overemphasizing way would be to clearly state the two different timescales involved sub (ps and μ s) and to write that the demonstrated properties might be utilized in the future for information processing.

We agree that as our present manuscript is rationalized on the dynamic response of ITO-NRA from a fundamental point of view, less emphasis on the signal switching and information processing aspect should help to give the manuscript a clearer focus. In the revised abstract, we have changed “thereby allowing for ultrafast signal switching and processing by optical means” to “which is potentially useful for all-optical information processing”.

Reviewer #3 (Remarks to the Author):

I reviewed the optical study of ITO nano-structure work done by Prof. Robert P. H. Chang group. The current paper carefully studied the optical transmittance of nano-structured ITO at visible regime. They performed both fast and slow transient studies and developed theoretical

modelling for the experimental explanations. They clearly demonstrated the capability to achieve transmittance signal modulation at visible frequencies up to 20% using the near-IR pumping induced plasmonic resonance effects. The data analysis is clear, the measurement results are convincing and their study is systematic and complete. I think this is really a good work and thus I recommend to publish after a revision.

Below there are several questions and comments I wish to ask:

It is known that metals are not an ideal platform for subwavelength light manipulation, and this becomes even obvious in the ultrafast time scale switching. The current developed metallic nanostructure certainly represents one strategy to overcome these issues. I also noticed that very recently, the authors had studied ultrafast tunability at mid-IR and near-IR regime (*Nature Photonics*, 10, 267 (2016)). Thus it is good to see a full picture over wide frequencies of this system. In the meantime, there also has other development using novel functional material, such as graphene and metamaterials for example (*Nature Photonics*, 10, 244 (2016); *Appl. Phys. Lett.* 104, 141102 (2014)). It would be great if the authors could cite and comment on these developments along this direction.

We thank the Reviewer for pointing out these highly relevant articles, which can provide a broader context for the present study. In the introduction section of the revised manuscript, we added (1) *Appl. Phys. Lett.*, 2014, **104**, 141102, (2) *Nat. Photonics*, 2016, **10**, 244-247, and (3) *Nat. Nanotechnol.*, 2015, **10**, 682-686, which now appear as ref. 5, 12 and 13, respectively. We also commented on these developments by adding the following sentences “Graphene as an emerging active plasmonic material has attracted significant attention as it offers ultrafast response in the sub-picosecond regime¹², and can be further interfaced with other two-dimensional atomic crystals for dispersion engineering¹³; however, the operation wavelength of graphene based plasmonics is primarily limited to the mid-infrared (MIR) range”.

The current reported maximum light manipulation in the visible regime is still less than 30%, can the authors comment on what are the limiting factors, this would be helpful for the general readers and possibly new works following the current study.

Our work has demonstrated absolute transmission modulation beyond $\pm 20\%$, and differential transmission up to 300% (with one of the two additional samples shown in Fig. 8 in the revised manuscript) in the visible range. The amplitude of the absolute modulation reached in our work, to our knowledge, is in fact higher than what have been previously demonstrated (such as *Nat. Photonics*, 2009, **3**, 55-58; *Nat. Nanotechnol.*, 2011, **6**, 107-111; *Nat. Commun.*, 2014, **5**, 4869; *Nat. Nanotechnol.*, 2015, **10**, 770-774) under the simple transmission or reflection configurations.

On the signal beam (probe) side, the modulation amplitude is partly limited by the relatively broad resonance (low quality factor) in the visible. Specially designed optical resonators such as ring resonators and disk resonators, which possess high quality factors (sharp transmission dips), may offer larger modulation amplitude (that may even be achieved

at low pumping fluences), although the footprint of light may need to be reduced and the spectral width of the modulation may be compromised. There is, for example, excellent discussion made in a recent publication *Nat. Photonics*, 2016, **10**, 227-238.

On the control beam (pump) side, in this work we only explored intraband pumping of the electron gas confined in the conduction band. However, as shown in *JACS*, 2015, **137**, 518-524, interband pumping of electrons from the valence band to the conduction band is also expected to alter the electron distribution and should result in a change of permittivity. This may potentially offer larger permittivity modulations.

In the revised manuscript, we added “Further improvements of the modulation intensity may be achievable by specially designed high quality-factor optical resonators⁴⁸ operating at a narrower bandwidth, or exploiting other pumping schemes such as interband pumping of electrons from valence to the conduction band⁴⁹” to the conclusion paragraph, and cited the reference *Nat. Photonics*, 2016, **10**, 227-238 and *JACS*, 2015, **137**, 518-524, appearing as reference 48 and 49, respectively.

Below are some of my detailed comments and suggestions to change:

1) One important aspect need to improve is that the transition from ΔT to $\Delta \epsilon$ is a bit abrupt, and it is not entirely clear how does the $\Delta \epsilon$ (both real and imag) are deduced and calculated. A detailed explanations and formula are recommended.

In our manuscript, we used two approaches to estimate the permittivity change.

The first approach is to combine the experimentally observed redshifts of transmission minima ($\Delta \lambda_m$, $m = 1$ to 5), and the waveguide simulation predicted relationship of $\Delta \epsilon'$ v.s. $\Delta \lambda_m$, to deduce $\Delta \epsilon'$ in our TA experiments as a function of both wavelength and pump fluence. The pump fluence can be further related to the electron temperature rise based on the procedure we described earlier in article *Nat. Photonics*, 2016, 10, 267. Note that this approach only gives quantitative information about $\Delta \epsilon'$ (the real part of permittivity) but not $\Delta \epsilon''$ (the imaginary part of permittivity); the result appeared in Fig. 3c and 5a in the revised manuscript.

The second approach is to theoretically calculate the electron temperature and wavelength dependent $\Delta \epsilon''$ and $\Delta \epsilon'$. The details (explanations and formula) were shown in the last section of the Supplementary Info, which we did not include as a section in the main text due to the limit on word counts and number of figures.

We recognize that the overall analysis could be complicated and the Reviewer's concern is well justified. To improve the transition from ΔT to $\Delta \epsilon$ (which is associated with the first approach), we explicitly defined the first approach as the “experimental approach”, and the second approach as the “theoretical approach”. In particular, in the revised manuscript we have added the following sentences “This approach of combining the experimentally observed redshifts of transmission minima ($\Delta \lambda_m$) and the waveguide simulation predicted relation of $\Delta \epsilon'$ versus λ_m to deduce the $\Delta \epsilon'(\omega)$ in our TA experiments is denoted as the

experimental approach; note that this approach does not provide quantitative information about $\Delta\varepsilon''(\omega)$ ” at the end of the second paragraph in the section “Fast, electron-dominated component”. We have also added “This approach, which gives quantitative information about both $\Delta\varepsilon'(\omega)$ and $\Delta\varepsilon''(\omega)$, is denoted as the theoretical approach; details of this approach are discussed in Supplementary section 9” into the third paragraph in the same section. After clearly defining these two approaches, we make reference to the two approaches throughout the manuscript.

2) The authors mentioned from time to time the finite element simulations, but I missed the details explanation for this part, it would be good to add this;

We have performed three types of finite-element simulations, which are the “waveguide simulation” from which we obtained the effective mode index of the nanorod, “optical simulation” which yield the transmission and reflection of the periodic nanorod array, and “heat-transfer simulation” that provides the temperature profile of the nanorod during the slow component. We now explicitly refer to these three simulations in the manuscript, and have added simulation details in the Method section, under “finite-element simulations”.

3) The hot electron temperature is estimated to be as high as 14000 K, how does this temperature compared with others, can the authors comment on why it is so high?

The very high electron temperature reached in ITO is due to a low electron heat capacity, a direct result of the low electron concentration, in comparison to the noble metal counterparts. More dedicated discussions were made in our earlier article *Nat. Photonics*, 2016, **10**, 267-273, in particular in Fig. S1 in the SI and the last paragraph on p. 271 of the main text. Here we thank the Reviewer for pointing this out. In our revised manuscript we added the following sentence “The one order of magnitude higher electron temperature reached in ITO-NRA, compared to the gold nanorod counterpart pumped at similar fluence⁸, is due to a much smaller electron heat capacity, which further results from the low electron concentration in comparison to gold”, to the third paragraph of the section “Fast, electron-dominated component”.

4) How does ITO NRA doping concentration would play a role in the current studies?

In our very first publication (*ACS Nano*, 2011, **5**, 9161-9170, in particular Fig. 2d), we have studied the impact of post-annealing (which adjusts the oxygen vacancy concentration and effectively change the conduction electron concentration) on the LSPR wavelength, therefore we do not attempt to include this subject in the present manuscript, in which we aim to focus on the dynamic response of the system. Adjusting the doping concentration, as suggested by the Reviewer, leads to a very interesting situation, as it affects the material dielectric dispersion in both the visible and infrared ranges, as well as the conduction electron configuration. Here we provide our preliminary thoughts, while keeping the mindset that fully addressing this question warrants further careful and in-depth study, in which similar approach as provided in the present manuscript may be applied.

Assume that the conduction electron concentration is decreased (as the present sample is already in the high doping regime with an electron concentration in the 10^{21} cm^{-3} , thereby reducing the electron concentration is more realistic), which will yield a smaller plasma frequency ω_p ; this would have the following implications.

First, the Drude formula $\epsilon(\omega) = \epsilon_\infty - \omega_p^2/(\omega^2 + i\gamma_p\omega)$ dictates that permittivity will increase in the visible range as ω_p decreases. This is evident by the refractive index plots for epitaxial ITO and In_2O_3 films obtained by fitting their ellipsometry data, with indium oxide being the low doping limit. Note that due to a smaller effective band gap of In_2O_3 , a Lorentz pole is also pulled to longer wavelength (in the figure below this is manifested as a hump at $\sim 330 \text{ nm}$). The increase of the real permittivity will increase the dielectric contrast between the nanorod and air, and tune the effective mode index n_{eff} of the nanorod, and with it the spectral locations of the transmission minima dictated by the equation $\lambda_m = 2h(n_{\text{eff}} - 1)/(2m - 1)$.

Second, the LSPR wavelength will be red-shifted. Accordingly, to achieve strong modulation the pump wavelength will need to be adjusted to match the LSPR wavelength.

Third, a smaller electron concentration is accompanied by a smaller electron heat capacity, therefore under the same pump fluence the electron temperature rise will be higher. In the meantime, the room temperature electron chemical potential (denoted as $\mu_{300\text{K}}$) gets closer to the conduction band minimum (CBM). Note that evident from Supplementary Fig. S13c in the revised manuscript, in the range between $\mu_{300\text{K}}$ and CBM, $\Delta\epsilon''$ is negative; whereas above $\mu_{300\text{K}}$, $\Delta\epsilon''$ is always positive. As a result, the frequency range for $\Delta\epsilon'' < 0$ will shrink, while the frequency range of $\Delta\epsilon'' > 0$ will expand (as the electron temperature rise is larger, therefore higher energy states are accessible by the electrons). The amplitude and line-shape of the permittivity modulation in the visible range depends on these two factors, and is not obvious unless a dedicated study is performed.

5) Eq.1 might be confusing to the readers. Both the effective index and transmission minima are all denoted by n.

In the revised manuscript, the notation for the effective mode index becomes n_{eff} , whereas the transmission minima are now numbered by m ($m = 1, 2, 3, 4, 5$).

6) There are some typos in the text files in the SI;

We have carefully proofread the main text as well as the SI to correct the typos.

REVIEWERS' COMMENTS:

Reviewer #1 (Remarks to the Author):

The Authors have amended the manuscript based on the comments of the reviewers. In my view, the comments have been addressed appropriately and the manuscript is suitable for publication.

Reviewer #2 (Remarks to the Author):

The authors have responded to all comments of all reviewers in a very reasonable way. Additionally, they have included new data and new references upon request.

It is my opinion that the work can be published in its present state in Nature Communications.

Reviewer #3 (Remarks to the Author):

The authors have carefully addressed all of my questions and comments, thus i recommend publish as it is.